# *Bifidobacteria* and Mucosal-Associated Invariant T (MAIT) Cells: A New Approach to Colorectal Cancer Prevention?

**Hüseyin Sancar Bozkurt [1],* and Eamonn M. M. Quigley [2]**

[1]   Clinic of Gastroenterology, Medical Faculty Internal Medicine, Maltepe University, Maltepe 34843, Turkey
[2]   Gastroenterology and Hepatology, Lynda K. and David M. Underwood Center for Digestive Disorders,
      Houston Methodist Hospital and Weill Cornell Medical College, Houston, TX 77062, USA;
      equigley@houstonmethodist.org
*    Correspondence: sancarb79@gmail.com; Tel.: +90-009-050-5448-2291

**Abstract:** Colorectal cancer is the most preventable form of cancer worldwide. The pathogenesis of colorectal cancer includes gut inflammation, genetic and microbial composition factors. İmpairment of the gut microbiota has been associated with development of colorectal cancer. The genus *Bifidobacterium* is an important component of the commensal gut microbiota. *Bifidobacteria* are considered to have important roles in multiple homeostatic functions: immunologic, hormonal and metabolic. Mucosal-associated invariant T cells (MAIT) are components of the immune system involved in protection against infectious pathogens and regulate the pathogenesis of various inflammatory diseases and, potentially, colorectal cancer. Engagement between *Bifidobacterium* and MAIT cells could exert a beneficial effect on colorectal cancer prevention and treatment.

**Keywords:** *Bifidobacterium*; MAIT cells; colorectal cancer

## 1. Introduction

Colorectal cancer (CRC) is the third most commonly diagnosed cancer in males and the second in females around the world. Approximately 1.8 million new cases of CRC were diagnosed in 2018 and accounted for approximately eight percent of all cancer deaths [1]. Incidence and mortality rates vary markedly worldwide. The commensal gut microbiota plays important roles in various systemic functions which include modulation of the immune system, modulation of neuro-hormonal activity, gut barrier and epithelial integrity. Immune dysregulation, dysbiosis and epithelial disruption contribute to carcinogenesis in CRC. Of these, the development of inflammation and alterations in the colonic microbiota are the two factors most closely associated with progression to CRC [2–4].

## 2. *Bifidobacteria* and Gut Inflammation

The human gut microbiota includes commensal, symbiotic, and harmful bacteria [5,6]. It was demonstrated that colon microbiota have anti-inflammatory and anti-oncogenic features and contribute to the immune, neuroendocrine, and metabolic homeostasis of the host [7,8]. The genus *Bifidobacterium* comprises Gram-positive, non-motile, often branched anaerobic bacteria and belongs to the phylum *Actinobacteria* [9]. *Bifidobacteria* are one of the dominant species in the human gut microbiota and are frequently used as probiotics [10]. *Bifidobacterium* species have immunological, neurohormonal, and anti-inflammatory effects (Figures 1 and 2) [9,11–14]. *B. animalis* subsp. *lactis* exerts the highest level of intracellular hydrogen peroxide resistance among *Bifidobacteria* and could, thereby, provide protection against reactive oxygen species [15]. Previous studies have reported that *Bifidobacteria*

differ from other colonic bacteria in their role in carbohydrate metabolism [9,16]. *Bifidobacteria* use the fructose-6-phosphate phosphoketolase pathway to ferment oligosaccharides and indigestible oligosaccharides ingested by the host are converted into short chain fatty acids (SCFAs), such as butyrate, propionate, and acetate which provide beneficial effects on gut immunity and inflammation [17]. *Bifidobacteria* are the main sources of SCFAs production, and they are used as probiotic ingredients in many foods [18,19].

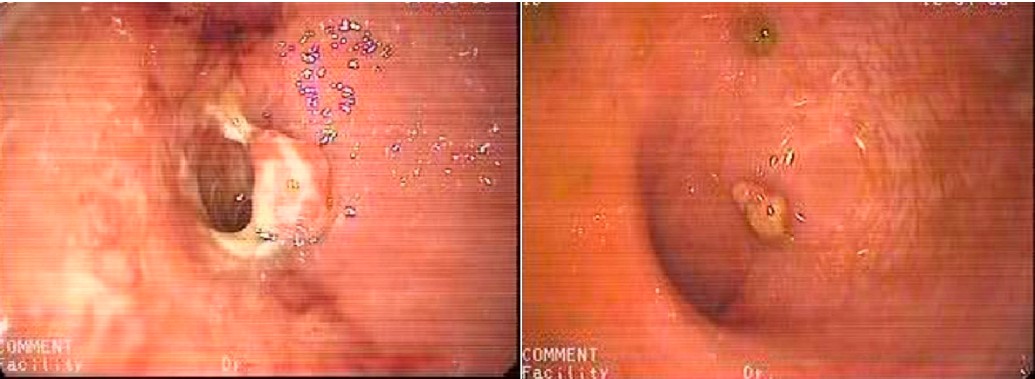

**Figure 1.** Mucosal healing (**right**) within one month after a single intracolonic application of 120 billion colony-forming units (CFUs) of *Bifidobacterium animalis* sp. *lactis* in unresponsive ulcerative colitis (**left**). From Hüseyin S. Bozkurt. 2019 [8] with permissions from Elsevier, Copyright 2019.

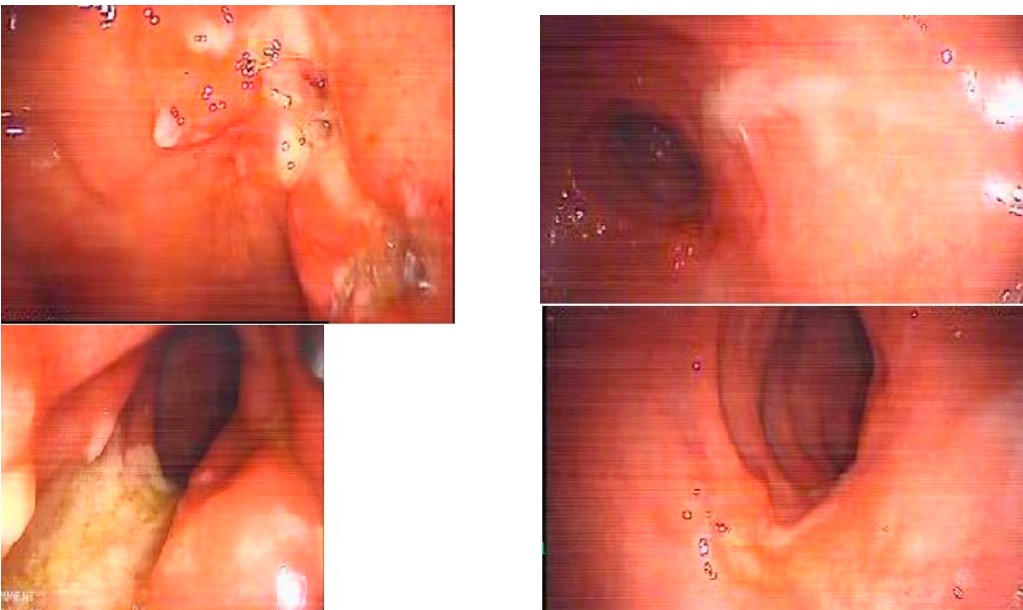

**Figure 2.** Mucosal healing (**right**) within one month after a single intracolonic application of 120 billion colony-forming units (CFUs) of *Bifidobacterium animalis* sp. *lactis* in unresponsive solitary rectal ulcer syndrome (**left**). From Hüseyin S. Bozkurt. 2019 [8] with permissions from Elsevier, Copyright 2019.

Reduced bifidobacterial levels are associated with inflammatory bowel disease (IBD) [20,21]. *Bifidobacteria* are able to protect against intestinal epithelial cell injury and this protection is independent of their effects on tumor necrosis factor alpha (TNF-α) production [20]. The exopolysaccharide (EPS) coat which is a feature of *Bifidobacteria* has been shown to play a significant role in this protective effect [22]. *Bifidobacteria* may also reduce cell injury as a direct result of inhibiting TNF-α and macrophages [20]. Also, *Bifidobacteria* increase regulatory T cell (Tregs) responses and, additionally, increase production of the anti-inflammatory cytokine interleukin-10 (IL-10) in IBD tissue [23]. In an

experimental model of IBD, *B. breve* ameliorated dextran sodium sulfate (DSS)-induced colitis. This was linked to increases in Tregs and decreases in CD4$^+$ IL-17$^+$ cells in Peyer's patches of DSS-induced colitis [24].

## 3. MAIT Cells and Gut Inflammation

Mucosal-associated invariant T (MAIT) cells are innate human T cells involved in antibacterial immunity. MAIT cells contain a T cell receptor (TCR) which has an invariant V$\alpha$7.2-J$\alpha$33 TCR$\alpha$ chain, and recognizes vitamin B precursors derived from microbial synthesis of riboflavin presented by the major histocompatibility complex (MHC) related 1 (MR1) [25,26]. Human MAIT cells are found in the lamina propria and other mucosal surfaces. MAIT cells are most abundant in the human gut lamina propria and associated organs, such as lymph nodes and the liver, also they can be found in peripheral blood (8% of all human T cells) [27]. MAIT cells were found to produce the Th1-related cytokines—interferon-gamma (IFN-$\gamma$) and TNF-$\alpha$ [28]. MAIT cells are stimulated by antigen-presenting cells infected by various species of bacteria. This stimulation requires the interaction between the invariant TCR and MR1; in this way, the activation of MAIT cells depends on the presence of the microbial flora. Previous studies have shown that MAIT cells might be implicated in IBD and CRC [29,30]. The expression of TCRV$\alpha$7.2 has been found to increase in inflamed colon tissue, while it has been shown to decrease in blood samples from IBD patients [31].

## 4. *Bifidobacteria* and Colon Cancer

Colonic epithelial inflammation is an important promoting factor in the development of colitis-related CRC. Gut inflammation occurs after mucosal invasion by colonic bacteria [32]. Later, persistent immune dysregulation and neoplastic changes arise in the colon mucosa. Chung et al. showed that *Bacteroides fragilis* promotes a pro-carcinogenic, inflammatory cascade that requires IL-17R and includes nuclear factor (NF)-$\kappa$B signaling in colonic epithelial cells in the context of intestinal dysbiosis [33]. İnvasion of the protective mucus layer of the colon by pathogenic bacteria causes colonic instability and DNA damage begins with neoplastic change accompanying chronic inflammation [34]. It was demonstrated that *Bifidobacterium breve* reduces the expression of IL-17(Th17) and IL-23, which play an important role in the development of IBD [35]. Schroeder et al. [36] reported that *Bifidobacterium longum* promotes mucosal integrity and corrects dysbiosis. Also, it has been shown that when colonic permeability is decreased, enhancement of the epithelial mucus layer is increased in the presence of a balanced colonic bacterial population [36]. Butyrate has strong anti-inflammatory and anti-tumor effects and *Bifidobacteria* are considered the main source of butyrate production and [37]. İt was shown that a higher diversity of butyrate-producing bacteria is found in stools of native Africans with low CRC risk as compared to Afro-Americans with a higher risk [38]. Also, it was determined that butyrate inhibits CRC cell proliferation and promotes differentiation and apoptosis of CRC cells [39]. Higher levels of butyrate production reduce the incidence of carcinogen-induced colon tumors. Free fatty acid receptor 2 (Ffar2) is a receptor for SCFAs and Ffar2 is downregulated in human colon cancers [40]. Sivaprakasam et al. reported that the administration of *Bifidobacterium* reduces intestinal inflammation and carcinogenesis in Ffar2$^{-/-}$ mice [40]. Butyrate may play an important role in oncogenesis, genomic instability, inflammation, and colon cell energy metabolism.

## 5. *Bifidobacteria* and MAIT Cells in Colon Cancer

MAIT cells are stimulated in IBD, and their accumulation in inflamed colon mucosa correlates with IBD activity. Kentaro et al. reported that the numbers of MAIT cells were significantly lower in the peripheral blood and significantly higher in inflamed colon tissue of IBD patients compared to healthy controls [29]. They used immunohistochemistry to examine MAIT cells in the inflamed mucosa of IBD patients and the normal mucosa of colon cancer patients. They showed that surgically resected colon specimens of patients with ulcerative colitis (UC) and colon cancer stained for anti-TCR-V$\alpha$7.2

antibodies. According to their findings, an accumulation of MAIT cells in the inflamed mucosa correlated with their decrease in the peripheral blood of IBD patients with more active disease.

Shaler et al. [41] found that CD3$\varepsilon$+V$\alpha$7.2+CD161++ MAIT cells infiltrated hepatic metastases in patients with colorectal carcinoma. Mansour Haeryfar et al. [42] hypothesized that MAIT cells could constitute attractive targets for cancer immunotherapy. Activated circulating MAIT cells from CRC patients promoted the production of IL-17 [43]. IL-17 could promote angiogenesis and cancer progression in CRC [43]. CRC usually disrupts mucosal homeostasis and barrier function. CRC development and progression depend on the interaction in the microenvironment around the tumor between neoplasia, pathogens, and tumor-infiltrating MAIT cells [44]. Arthur et al. reported that intestinal inflammation promoted by MAIT cells can alters the prognosis for the tumor and microbial composition [45,46]. They demonstrated that microbial composition modulates the progression of colitis-associated CRC using the colitis-susceptible *Il10*$^{-/-}$ mouse strain [45,47] and they also reported that inflammation, rather than cancer, was related to colonic microbial shifts. MAIT cells can affect clinical outcome and survival of CRC patients [48]. Tosolini et al. [48] showed that CRC patients exhibiting high expression of the Th17 (IL-17) cluster had a poor prognosis, whereas patients with high expression of the Th1 cluster had prolonged disease-free survival. They determined that functional Th1 and Th17 clusters exert opposite effects on patient survival in CRC and, interestingly, none of the Th2 clusters (IL4, IL5, IL13) were predictive of prognosis. Rui Yu et al. showed that *Bifidobacterium adolescentis* strains were associated with the induction of Th17 cells in humans [49]. Also, Ruiz et al. reported that *Bifidobacterium animalis* strains promote a Th1 response, in both in vitro and in vivo experiments [50]. Wei et al. reported that the use of transfected *Bifidobacteria* as a novel system to induce specific genes offered a promising therapeutic approach to treat a tumor through non-stimulatory effects on Th17 cells [51]. In summary, it seems likely that MAIT Th17 cells preferentially infiltrate into the tumor in CRC patients and may contribute to prognosis of CRC.

MAIT-TCRs levels stimulated by colonic antigen-presenting cells were measured in an in vitro assay and the stimulatory effects of 47 microbiota-associated bacterial strains from different phyla assessed [52]. Most species that are high-stimulators for MAIT-TCRs belong to the *Bacteroidetes* and *Proteobacteria phyla*, whereas low/non-stimulator species belong to the *Actinobacteria* or *Firmicutes phyla*. Also, riboflavin metabolites from high and low MAIT-stimulating bacteria that possessed the riboflavin pathway were measured. Interestingly, it was reported that human T cell subsets can also present riboflavin metabolites to MAIT cells in an MR1-restricted fashion and this signaling also contributed to increased production of IFN-$\gamma$ and TNF-$\alpha$ [52].

It appears that the *Bifidobacterium animalis* strain exhibits low/non-stimulator status for MAIT cells and it can be proposed that the *Bifidobacterium animalis* strain may be effective in preventing CRC through non-stimulatory effects on Th17 (IL17) cells and a promoting effect on Th1 cells.

## 6. Conclusions

Significant progress development has been made in recent years in recognizing the importance of the interaction between the gut microbiota and MAIT cells in CRC. *Bifidobacterium* strains play protective and preventive roles on human colonic microbiota composition and may have an impact on the inflammatory regulation of CRC. *Bifidobacterium* strains may be effective in preventing CRC through their inhibitory effects on MAIT cells.

**Author Contributions:** Writing—original draft, H.S.B.; Writing—review & editing, E.M.M.Q.

**Funding:** This research received no external funding.

**Conflicts of Interest:** The authors declare no conflict of interest.

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
