# Peer review of "Bifidobacteria and Mucosal-Associated Invariant T (MAIT) Cells: A New Approach to Colorectal Cancer Prevention?"

_gastrointestdisord, doi:10.3390/gidisord1020022_

Round 1
Reviewer 1 Report
The manuscript is quite well written.
The methods are adequate.
The results justify the conclusions drawn.
It would be useful for the readers to include the discussion of PMID:30646431
Author Response
1- English language has been reedited
2- PMID:30646431 has been cited in the discussion text(Red Highlighted)

Reviewer 2 Report
The paper is interesting.
I have onequestion regarding this paper:
I suggest the authors to read this manuscript “Yu R, Zuo F, Ma H, Chen S. Exopolysaccharide-Producing Bifidobacterium adolescentis Strains with Similar Adhesion Property Induce Differential Regulation of Inflammatory Immune Response in Treg/Th17 Axis of DSS-Colitis Mice. Nutrients. 2019 Apr 4;11(4)” because is interesting for this review
Author Response
Dear Sir,
1- English language has been reedited
2-"Yu R, Zuo F, Ma H, Chen S. Exopolysaccharide-Producing Bifidobacterium adolescentis Strains with Similar Adhesion Property Induce Differential Regulation of Inflammatory Immune Response in Treg/Th17 Axis of DSS-Colitis Mice. Nutrients. 2019 Apr 4;11(4)" has been cited int he discussion text(Red Highlighted)

Reviewer 3 Report
In this review authors explored the role of Bifidobacteria and Mucosal-Associated Invariant T (MAIT) cells in the prevention of colorectal cancer. Even though this topic is inquisitive, the evidence presented here is not comprehensive enough for its acceptance.
Author Response
Dear Sir,
1-English language has been reedited
2-The role of Bifidobacteria and Mucosal-Associated Invariant T (MAIT) cells in the prevention of colorectal cancer has been recited and reedited extensively in the discussion (Red highlighted)

Reviewer 4 Report
The authors present and interesting topic here on Bifidobacteria and Mucosal-Associated Invariant T (MAIT) cells: A new approach to colorectal cancer prevention?
Review article is an interesting read. Few corrections will need to be made prior to acceptance of the article.
Line 26 after the sentence “account for approximately 8 percent of all cancer deaths [1]” I would also include the latest review article “Rawla P, Sunkara T, Barsouk A. Epidemiology of colorectal cancer: incidence, mortality, survival, and risk factors. Gastroenterology Review/Przegląd Gastroenterologiczny. 2019. doi:10.5114/pg.2018.81072.” as a reference. Please mention about the different numbers for incidence and mortality for colorectal carcinoma.
In line 36 at the end of the sentence please include the citation of the article on FMT by Sunkara et al PMID: 30214266
In Bifidobacterium and Colon cancer paragraph please include few lines about Future treatment prospects using bifidobacterial. Please include the citation by Wei et al PMID: 29963126.
Overall a very nicely written review article.
Author Response
Dear Sir,
Red Highlighted on text:
1-English Language has been reedited
2-İntroduction has been reedited
3-Line 26 sentence has been reedited and recited
4- PMID: 30214266 has been cited in Line 36 sentence
5- Wei et al PMID: 29963126 has been cited as a bifidobacterial using in tumor in the discussion text

Round 2
Reviewer 3 Report
The revised version looks much better. I recommend for acceptance.